

# An observational descriptive survey of rosacea in the Chinese population: clinical features based on the affected locations

Hong-fu Xie[1,*], Ying-xue Huang[1,*], Lin He[1], Sai Yang[1], Yu-xuan Deng[1], Dan Jian[1], Wei Shi[1] and Ji Li[1,2]

[1] Department of Dermatology, Xiangya Hospital, Central South University, Changsha, China
[2] Key Laboratory of Organ Injury, Aging and Regenerative Medicine of Hunan Province, Changsha, China
[*] These authors contributed equally to this work.

## ABSTRACT

**Background**. There is currently no study that has evaluated the differences in epidemiological and clinical characteristics among rosacea patients according to different facial sites.

**Methods**. Clinical and demographic data were obtained from 586 rosacea patients. The patients were divided into four groups based on the main sites involved with the rosacea lesions (full-face, cheeks, nose, or perioral involvement). Clinical signs were measured through self-reported, dermatologist-evaluated grading of symptoms, and physiological indicators of epidermal barrier function.

**Results**. There were 471 (80.4%), 49 (8.4%), 52 (8.9%), and 14 (2.4%) cases in the full-face, cheek, nasal and perioral groups, respectively. Compared with the healthy control, the full-face group had lower water content and higher transepidermal water loss (TEWL) in the cheeks, and chin; the perioral group had lower water content and higher TEWL in the chin; while the nasal group had the normal water content and TEWL. Compared with the full-face group, the nasal group had more severe phymatous changes, less severe self-reported and dermatologist-evaluated grading of symptoms. All the patients in the perioral or the nasal group had their first rosacea lesions start and remain at the chin or on the nose. In the full-face group, 55.8% of patients had their lesions start with the full face, 40.1% on the cheek, and the rest (4.1%) on the nose.

**Conclusion**. Significant differences in clinical features were observed among rosacea patients with lesions at four different sites. The lesion localization of each group was relatively stable and barely transferred to other locations.

Corresponding author
Ji Li, liji0704@163.com, lydia.1208@hotmail.com

## INTRODUCTION

Rosacea is a common chronic inflammatory cutaneous disorder, predominantly presenting on the faces of adults, which is characterized by a tendency of frequent facial flushing, central facial erythema, papulopustules, telangiectasias, ocular manifestations, and phymatous

changes primarily on the nose. The exact cause of rosacea remains unclear. It may contain hereditary components, and has been hypothesized to be associated with disorders of the innate immune system, dysfunction of facial vascular regulation, neurogenic inflammation, and elevated levels of Demodex mites, among others (*Gibson, 2004*; *Yamasaki & Gallo, 2009*; *Abram et al., 2010*; *Steinhoff et al., 2011*; *Van Zuuren et al., 2015*; *Margalit et al., 2016*). Morbidity of rosacea varies greatly among different ethnic populations, with a higher prevalence amongst fair-skinned individuals of northern European or Celtic ancestry (*Spoendlin et al., 2012*; *Tuzun et al., 2014*). There has yet to be any published data regarding the incidence of rosacea in the Chinese population.

The current classification system for rosacea describes four distinct clinical subtypes: erythematotelangiectatic, papulopustular, phymatous, and ocular (*Wilkin et al., 2002*). Erythematotelangiectatic rosacea is characterized by flushing and persistent central facial erythema with or without telangiectasia. Papulopustular rosacea is associated with persistent central facial erythema with transient, central facial papules or pustules or both. Phymatous rosacea is characterized by skin thickening, irregular surface nodularities, and enlargement, which can affect the nose, chin, forehead, ears, and eyelids. The most commonly affected area is the nose, which is also called rhinophyma. There are three grades of phymatous changes: (1) mild, manifested as puffiness and mildly patulous follicles with no clinically apparent hypertrophy of connective tissue or sebaceous glands and no change in contour; (2) moderate, manifests as moderate swelling and moderately dilated patulous follicles with clinically mild hypertrophic change in nasal contour but no nodular components; (3) severe, manifests as marked swelling and large dilated follicles with distortion of contour with a nodular component. Ocular rosacea is a subtype that displays a series of non-specific ocular symptoms. For the first three subtypes, the present method of assessing the severity of this disease classifies the progression of rosacea into four general stages (*Wilkin et al., 2004*): Stage 1, which is characterized by frequent blushing; Stage 2, which is characterized by transient erythema of the central areas of the face, and obvious, but slight, telangiectasias; Stage 3, which includes more severe facial erythema, increased telangiectasias, and papule and pustule formation; and Stage 4, which is the most severe, and is also known as rhinophyma (*Zuber, 2000*). Based on this classification system, it could be inferred that the stages of rosacea might evolve from one to another and rhinophyma seemed to be the "end-stage" (*Wilkin, 1994*; *Jansen & Plewig, 1997*). A number of studies have also confirmed the possibility of progression between subtypes in western countries (*Crawford, Pelle & James, 2004*; *Powell, 2005*; *Tan & Berg, 2013*; *Tan et al., 2013*), but this theory is still being questioned (*Crawford, Pelle & James, 2004*). Moreover, during the progression process, patients can display a number of subtypes simultaneously, which makes the classification of rosacea vague and indistinct. Nevertheless, no other classification standards can define the clinical parameters more scientifically and reasonably. During our clinic work, we noticed some interesting phenomenon in the Chinese rosacea patients. For example, the clinical features varied among patients with different affected areas. Patients whose lesions first appeared on the nose could easily develop rhinophyma with thickened and nodular skin. While patients whose initial lesions occurred outside the nose area rarely progressed to clinically apparent hypertrophic changes of rhinophyma. In addition, based on our

preliminary observation, we supposed that the involved location might be the possible sign of the natural development of rosacea. However, few studies have evaluated the differences in clinical features and disease outcomes among rosacea patients on the basis of involved locations so far.

In our present descriptive survey, we recruited 586 rosacea patients from south of China, aiming to evaluate and compare the clinical features of rosacea at different sites of the face, assess quantitative details regarding the rosacea-associated symptoms, signs and indicators of epidermal barrier function, and analyze the potential for progression among different affected areas in the Chinese population, hoping to provide some evidences for future investigations on a better classification scheme.

## PATIENTS AND METHODS

### Patients

A total of 586 patients meeting the standard classification criteria for rosacea, as determined by the National Rosacea Society Expert Committee on the Classification and Staging of Rosacea (*Wilkin et al., 2002*), were admitted to the XiangYa Hospital from March 2013 to October 2014 and enrolled in the study consecutively after providing written informed consent. Exclusion criteria consisted of diseases or symptoms interfering with the evaluation, such as erosion, exudation, severe bacterial or fungal infection, and other skin diseases, pregnancy, lactation, pediatric cases and history of systematic disease. A total of 115 healthy individuals, who were all unselected volunteers, without history of rosacea or other diseases, were included as a healthy control group. Data input, organization, and analysis were conducted during May 2014 to February 2015. Authors had access to information that could identify individual participants during or after data collection. The study protocol was approved by the ethics review board of the XiangYa Hospital (Ethical Application Ref: 201212079), and the study was conducted in accordance with the Declaration of Helsinki. Informed written consents have been obtained from every participant.

### Methods

As shown in Fig. 1, the face can be divided into five main parts: the forehead, eyes, cheeks, nose, and perioral area. In this study, the pattern of skin involvement was classified into four groups: (1) full-face group (rosacea lesion occupied no less than two parts of the face); (2) cheek group (rosacea lesions limited to the cheeks); (3) nasal group (rosacea lesions limited to the nose); (4) perioral group (rosacea lesions limited to the perioral area).

Demographic and clinical data, including patient age, sex, disease duration, and self-reported symptoms, including burning, dryness, itching, stinging, skin tension, swelling, ants line sense, and pain (graded from 0 to 10, with 0 representing no symptoms and 10 representing most severe symptoms) were recorded by the patients. Moreover, three dermatologists graded the rosacea-associated symptoms based on the National Rosacea Society's grading system independently (0–3).
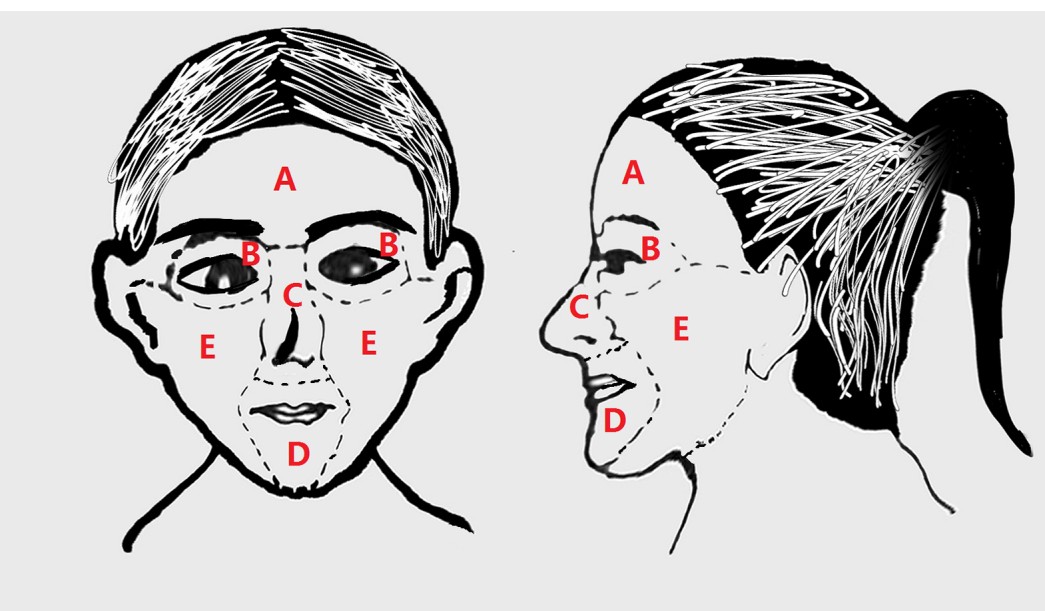

**Figure 1** **The face can be divided into five main parts: forehead, eyes, nose, perioral area, and cheeks, which can all be affected by rosacea.** (A) Forehead, (B) eyes, (C) nose, (D) perioral area, (E) cheeks.

## Physiological indicators of epidermal barrier function

Physiological indicators of epidermal barrier function, including skin water content, oil content, melanin, hemoglobin, transepidermal water loss (TEWL), and pH, were measured by Skin analysis SHP88 (Courage + Khazaka electronic GmbH, Germany) on the forehead, chin, cheeks, and nose. The same person conducted all the tests.

## Data analysis

Patient clinico-demographic characteristics were compared among all groups using the Pearson Chi-square test for categorical variables; Kruskal–Wallis test followed by all pair-wise multiple comparisons for ranked variables; and $t$ test or One-way ANOVA analysis with LSD multiple comparison for continuous variables. All reported $P$-values are two sided with $\alpha = 0.05$. All statistical analyses were performed using SPSS version 19.0 (SPSS Inc., Chicago, IL, USA).

## RESULTS

### Demographics

A total of 586 rosacea patients were included in this study: 501 (85.5%) women and 85(14.5%) men with a mean age of 32.7 years-old, and a median disease duration of 36 months. 115 healthy volunteers were also included in this study: 89 (77.4%) women and 26 (22.6%) men with a mean age of 37.2 years. The study group and the healthy control group did not show any significant differences with respect to sex and age ($P > 0.05$ for both, Chi-square test for sex and $t$ test for age). There were 471 (80.4%), 49 (8.4%), 52 (8.9%), and 14 (2.4%) cases in the full-face, cheek, nasal and perioral groups, respectively. The full-face group was consisted by 164 (34.8%) cases with lesions on the forehead, cheeks,

**Table 1  Patient demographics.**

| Demographic features | | Full-face group $n = 471$ (80.4%) | Cheek group $n = 49$ (8.4%) | Nasal group $n = 52$ (8.9%) | Perioral group $n = 14$ (2.4%) | Health control $n = 115$ |
|---|---|---|---|---|---|---|
| Sex, $n$ (%) | Female | 421 (89.4)[a] | 44 (89.8)[a] | 23 (44.2) | 13 (92.9)[a] | 89 (77.4) |
|  | Male | 50 (10.6)[a] | 5 (10.2)[a] | 29 (55.8) | 1 (7.1)[a] | 26 (22.6) |
| Age (years) | Mean ± SD | 32.4 ± 10.9 | 34.7 ± 11.2 | 33.8 ± 13.0 | 33.4 ± 11.3 | 37.2 ± 10.8 |
|  | Range | 13.0–66.0 | 15.0–58.0 | 15.0–80.0 | 16.0–47.0 | 20.0–50.0 |
| Duration (months) | Mean ± SD | 53.7 ± 58.3 | 53.3 ± 50.3 | 18.9 ± 13.2 | 53.2 ± 70.0 |  |
|  | Median | 36.0 | 36.0 | 12.0 | 30.0 |  |

**Notes.**

Abbreviations: SD, standard deviation.

[a] $P < 0.05$, compared with the nasal group, Chi-square test.

nasal and perioral area; 39 (8.3%) on the forehead, cheeks and nasal area; 82 (17.4%) on the forehead and cheeks, 48 (10.2%) on the cheeks, nasal and perioral area; 36 (7.64%) on the cheeks and perioral area; 73 (15.5%) cases with lesions on the forehead, cheeks, and perioral area; and 29 (6.16%) on the cheek and nasal area in the full-face group, respectively. The proportion of men was higher in the nasal group than in the full-face, the cheek and perioral groups ($P < 0.001$ for all, Chi-square test), whereas the mean age and disease duration were both similar for each patient groups ($P > 0.05$ for both, ANOVA). All were shown in Table 1.

## Epidermal barrier function

The physiological indicators of epidermal barrier function of rosacea in the different groups are shown in Table 2. The nasal group had higher oil content on the nose than the healthy control and full-face group ($P = 0.007$ and $P = 0.039$, ANOVA, LSD Multiple comparison), and had higher hemoglobin in the nose than the healthy control and full-face group ($P < 0.001$ for both, ANOVA, LSD Multiple comparison). Also, it showed higher melanin in the forehead, cheeks and chin than the other groups ($P < 0.05$ for all, ANOVA, LSD Multiple comparison). The full-face group had lower water content in the cheek, nose and chin compared to that of the healthy control ($P = 0.047$, $P = 0.032$ and $P = 0.041$, ANOVA, LSD Multiple comparison), but had higher hemoglobin in the forehead, cheeks and chin than that of the healthy control ($P < 0.05$ for all, ANOVA, LSD Multiple comparison), and higher hemoglobin in the cheeks than that of the nasal group ($P < 0.001$, ANOVA, LSD Multiple comparison). All the four patient groups had higher hemoglobin in the cheeks compared to that of the healthy control ($P < 0.05$ for all, ANOVA, LSD Multiple comparison). The perioral group had the lowest water content in the chin ($P < 0.001$ for all, ANOVA, LSD Multiple comparison); it also had higher hemoglobin in the chin compared to the healthy control group ($P = 0.006$, ANOVA, LSD Multiple comparison). The full-face group had higher TEWL compared to the healthy control and nasal group in the cheeks ($P = 0.039$ and $P = 0.005$, ANOVA, LSD Multiple comparison) and chin ($P = 0.000$ and $P = 0.021$, ANOVA, LSD Multiple comparison). The cheek groups had higher TEWL in the cheek than the nasal group ($P = 0.017$, ANOVA, LSD Multiple comparison). Moreover, the perioral group displayed higher TEWL in the
**Table 2 Physiological indicators of epidermal barrier function in different sites.**

| | Full-face group | Cheek group | Nasal group | Perioral group | Health control |
|---|---|---|---|---|---|
| **Forehead** | | | | | |
| Oil content ($\mu$g m$^{-2}$) | 70.98 ± 51.45 | 66.82 ± 52.49 | 87.79 ± 61.94 | 88.63 ± 46.93 | 55.55 ± 10.30 |
| Water content | 51.09 ± 14.19 | 50.59 ± 16.88 | 52.23 ± 12.95 | 55.38 ± 12.30 | 58.52 ± 7.08 |
| Melanin | 183.81 ± 41.28 | 178.37 ± 37.88 | 233.30 ± 69.09 | 204.84 ± 51.93 | 192.74 ± 32.11 |
| Haemoglobin | 407.87 ± 76.97[a] | 376.82 ± 56.36 | 393.77 ± 89.48 | 383.31 ± 60.72 | 342.62 ± 69.99 |
| TEWL (g m$^{-2}$ h$^{-1}$) | 8.21 ± 9.92 | 9.22 ± 6.58 | 5.04 ± 2.79 | 6.72 ± 2.31 | 5.24 ± 1.82 |
| pH | 5.16 ± 0.65 | 5.22 ± 0.29 | 5.14 ± 0.41 | 5.44 ± 0.44 | 5.11 ± 0.40 |
| **Cheek** | | | | | |
| Oil content ($\mu$g m$^{-2}$) | 46.89 ± 40.08 | 39.88 ± 30.20 | 53.83 ± 41.37 | 60.45 ± 44.11 | 40.11 ± 16.40 |
| Water content | 50.59 ± 16.32[a] | 47.64 ± 20.11 | 53.32 ± 16.27 | 55.56 ± 12.21 | 61.74 ± 14.27 |
| Melanin | 156.60 ± 48.20[b] | 137.27 ± 53.2[b] | 189.95 ± 44.68 | 161.39 ± 39.43[b] | 147.22 ± 31.84[b] |
| Haemoglobin | 439.96 ± 81.9[a,b] | 408.52 ± 97.56[a] | 360.03 ± 95.04[a] | 424.90 ± 77.42[a] | 277.07 ± 52.67 |
| TEWL (g m$^{-2}$ h$^{-1}$) | 9.99 ± 9.95[a,b] | 10.47 ± 7.68[b] | 5.12 ± 2.81 | 6.74 ± 2.52 | 6.05 ± 4.67 |
| pH | 5.33 ± 0.64 | 5.41 ± 0.29 | 5.12 ± 0.34 | 5.59 ± 0.33 | 5.24 ± 0.42 |
| **Nose** | | | | | |
| Oil content ($\mu$g m$^{-2}$) | 94.06 ± 63.22[b] | 112.44 ± 68.64 | 116.55 ± 65.72 | 81.25 ± 23.07 | 57.11 ± 18.82[b] |
| Water content | 41.55 ± 17.39[a] | 45.95 ± 12.24 | 45.88 ± 13.57 | 49.12 ± 9.24 | 53.77 ± 9.35 |
| Melanin | 214.86 ± 42.61[b] | 215.26 ± 34.59[b] | 235.51 ± 63.60 | 206.25 ± 30.12[b] | 203.60 ± 36.51[b] |
| Haemoglobin | 442.36 ± 87.68[b] | 443.00 ± 79.41 | 547.51 ± 116.68 | 463.9 ± 37.57 | 373.77 ± 62.84[b] |
| TEWL (g m$^{-2}$ h$^{-1}$) | 7.52 ± 5.59 | 6.50 ± 3.29 | 7.63 ± 4.22 | 6.92 ± 2.06 | 5.20 ± 2.31 |
| pH | 5.21 ± 0.65 | 5.00 ± 0.27 | 5.06 ± 0.23 | 5.38 ± 0.3 | 5.02 ± 0.35 |
| **Chin** | | | | | |
| Oil content ($\mu$g m$^{-2}$) | 71.24 ± 51.02 | 60.11 ± 44.16 | 86.91 ± 64.18 | 66.45 ± 32.01 | 48.77 ± 10.83 |
| Water content | 56.10 ± 12.42[a] | 57.81 ± 13.10 | 52.58 ± 15.15 | 38.38 ± 10.34[d] | 64.95 ± 13.60 |
| Melanin | 214.46 ± 48.21[b] | 207.87 ± 51.72[b] | 259.45 ± 57.92 | 217.60 ± 59.05[b] | 215.55 ± 44.56[b] |
| Haemoglobin | 487.01 ± 89.49[a] | 459.36 ± 64.45 | 464.47 ± 85.28 | 520.31 ± 67.08[a] | 412.92 ± 74.72 |
| TEWL (g m$^{-2}$ h$^{-1}$) | 9.57 ± 5.61[a,b] | 10.58 ± 6.26 | 6.93 ± 3.54 | 14.03 ± 5.50[a,b,c] | 4.78 ± 3.03 |
| pH | 5.25 ± 0.64 | 5.24 ± 0.32 | 5.06 ± 0.29 | 5.48 ± 0.35 | 5.01 ± 0.52 |

**Notes.**

Abbreviations: TEWL, transepidermal water loss.

[a] $P < 0.05$, compared with the health control group, ANOVA, LSD Multiple comparison.
[b] $P < 0.05$, compared with the nasal group, ANOVA, LSD Multiple comparison.
[c] $P < 0.05$, compared with the full face group, ANOVA, LSD Multiple comparison.
[d] $P < 0.05$, compared with the other groups, ANOVA, LSD Multiple comparison.

chin compared to the healthy control, full-face and nasal group ($P = 0.01$, $P = 0.006$ and $P = 0.000$, ANOVA, LSD Multiple comparison), but no significant differences in TEWL in the forehead and nose was observed among the five groups ($P = 0.965$, ANOVA).

## Self-reported symptoms

As shown in Table 3, the nasal group, compared with the other patient groups, had less severe burning, drying, itching, stinging, skin tension, self-reported symptoms ($P < 0.05$ for all, Kruskal–Wallis test followed by all pair-wise multiple comparisons). The perioral group, compared with the full-face group, had more severe swelling ($P = 0.04$, Kruskal–Wallis test followed by all pair-wise multiple comparisons). There was no difference in severity of other

**Table 3   Values of self-reported symptoms in the different groups.**

|  | Full-face group | Nasal group | Perioral group | Cheek group |
|---|---|---|---|---|
| **Burning** | | [a] | | |
| Median | 5.0 | 0.0 | 3.0 | 5.0 |
| Lower-upper quartile | 2.0–7.0 | 0.0–3.0 | 0.0–3.0 | 1.0–7.0 |
| **Dry** | | [a] | | |
| Median | 4.0 | 0.0 | 5.0 | 4.0 |
| Lower-upper quartile | 1.0–6.0 | 0.0–1.8 | 2.0–6.0 | 1.0–7.0 |
| **Itching** | | [a] | | |
| Median | 3.0 | 0.5 | 3.0 | 3.0 |
| Lower-upper quartile | 1.0–5.0 | 0.0–3.0 | 1.0–6.0 | 1.0–5.5 |
| **Stinging** | | [a] | | |
| Median | 0.0 | 0.0 | 0.0 | 0.0 |
| Lower-upper quartile | 0.0–3.0 | 0.0–0.0 | 0.0–5.0 | 0.0–1.0 |
| **Skin tension** | | [a] | | |
| Median | 3.0 | 0.0 | 0.0 | 1.0 |
| Lower-upper quartile | 0.0–5.0 | 0.0–0.0 | 0.0–3.0 | 0.0–5.5 |
| **Swelling** | | | [b] | |
| Median | 0.0 | 0.0 | 3.0 | 0.0 |
| Lower-upper quartile | 0.0–3.0 | 0.0–2.0 | 0.0–5.0 | 0.0–1.0 |
| **Ant line sense** | | | | |
| Median | 0.0 | 0.0 | 0.0 | 0.0 |
| Lower-upper quartile | 0.0–2.0 | 0.0–0.0 | 0.0–4.0 | 0.0–1.0 |
| **Pain** | | | | |
| Median | 0.0 | 0.0 | 0.0 | 0.0 |
| Lower-upper quartile | 0.0–0.0 | 0.0–0.8 | 0.0–3.0 | 0.0–1.0 |
| **Overall symptoms** | | [a] | | |
| Median | 18.0 | 4.5 | 22.0 | 17.0 |
| Lower-upper quartile | 11.0–28.0 | 0.0–11.8 | 13.0–27.0 | 11.0–24.0 |

**Notes.**
[a] $P < 0.05$, compared with the other groups, Kruskal–Wallis test followed by all pair-wise multiple comparisons.
[b] $P < 0.05$, compared with the full face group, Kruskal–Wallis test followed by all pair-wise multiple comparisons.

self-reported symptoms among the full-face, cheek, and the perioral groups ($P > 0.05$, Kruskal–Wallis rank sum test).

## Doctor-evaluated grading severity
### Primary features

The frequency of primary rosacea features in different groups is shown in Table 4. In total, 514 (87.7%) patients reported flushing with the following severity distribution: absent (12.3%), mild (27.3%), moderate (20.1%), severe (40.3%). The full-face group or cheek group is more likely to have flushing than the nasal or perioral group (91.3% and 91.8% vs. 57.7% and 64.3%, $p < 0.05$, Chi-square test). Although the frequency of flushing between the full-face and the cheek group was similar, the full-face had the highest severity of flushing among the patient groups ($P < 0.05$, Kruskal–Wallis test followed by all pair-wise

**Table 4  Frequency of primary rosacea features in the different groups.**

| | Full-face group | Cheek group | Nasal group | Perioral group | Total |
|---|---|---|---|---|---|
| **Flushing, *n* (%)** | [f] | [a] | | | |
| Absent | 41 (9)[b,c] | 4 (8)[b,c] | 22 (42) | 5 (36) | 72 (12.3) |
| Mild | 117 (25) | 23 (47) | 15 (29) | 5 (36) | 160 (27.3) |
| Moderate | 102 (22) | 9 (18) | 6 (12) | 1 (7) | 118 (20.1) |
| Severe | 211 (44) | 13 (27) | 9 (17) | 3 (21) | 236 (40.3) |
| **Non-transient erythema, *n* (%)** | | | | | |
| Absent | 17 (4) | 4 (8) | 8 (15) | 0 (0) | 29 (5) |
| Mild | 308 (65) | 33 (67) | 26 (50) | 10 (71) | 377 (64.3) |
| Moderate | 113 (24) | 10 (20) | 8 (15) | 4 (29) | 135 (23) |
| Severe | 33 (7) | 2 (5) | 10 (20) | 0 (0) | 45 (7.7) |
| **Papulopustules, *n* (%)** | [a,e] | | | [a] | |
| Absent | 63 (13) | 20 (41) | 19 (37) | 1 (7) | 103 (17.6) |
| Mild | 127 (27) | 10 (20) | 22 (42) | 4 (29) | 163 (27.8) |
| Moderate | 106 (23) | 12 (25) | 9 (17) | 8 (57) | 135 (23) |
| Severe | 175 (37) | 7 (14) | 2 (4) | 1 (7) | 185 (31.6) |
| **Telangiectasia, *n* (%)** | [a,d] | | | | |
| Absent | 96 (20) | 9 (18) | 20 (39) | 5 (36) | 130 (22.2) |
| Mild | 113 (24) | 17 (35) | 14 (27) | 6 (43) | 150 (25.6) |
| Moderate | 108 (23) | 13 (27) | 8 (15) | 3 (21) | 132 (22.5) |
| Severe | 154 (33) | 10 (20) | 10 (19) | 0 (0) | 174 (29.7) |

**Notes.**
[a] $P < 0.05$, compared with the nasal group, Kruskal–Wallis test followed by all pair-wise multiple comparisons.
[b] $P < 0.05$, compared with the nasal group, Chi-square test.
[c] $P < 0.05$, compared with the perioral group, Chi-square test.
[d] $P < 0.05$, compared with the perioral group, Kruskal–Wallis test followed by all pair-wise multiple comparisons.
[e] $P < 0.05$, compared with the cheek group, Kruskal–Wallis test followed by all pair-wise multiple comparisons.
[f] all $P < 0.05$, compared with the other groups, Kruskal–Wallis test followed by all pair-wise multiple comparisons.

multiple comparisons). The cheek group also had more severe flushing than the nasal group ($P = 0.02$, Kruskal–Wallis test followed by all pair-wise multiple comparisons).

Non-transient erythema manifested in 95.1% of the patients, namely the full-face group (96.4%), cheek group (91.8%), nasal group (84.6%), and perioral group (100%). There was no significant difference in the severity distribution of non-transient erythema among patient groups ($P > 0.05$, Kruskal–Wallis rank sum test).

Papulopustules were present in 92.9% of the perioral group, 86.6% of the full-face group, 62.7% of the nasal group, and 59.2% of the cheek group. The full-face had more severe papulopustules than the nasal group and the cheek group ($p < 0.001$ for both, Kruskal–Wallis test followed by all pair-wise multiple comparisons) and papulopustules were more severe in the perioral group than in the nasal group ($p = 0.002$, Kruskal–Wallis test followed by all pair-wise multiple comparisons).

Telangiectasia was present in all the patient groups: full-face group (79.6%), cheek group (81.6%), nasal group (61.5%), and perioral group (64.3%). There was no significant difference in the severity of telangiectasia between the nosal, cheek, and perioral groups. The full-face group had more severe telangiectasia than the nasal or perioral group ($p = 0.002$

and $p = 0.007$, Kruskal–Wallis test followed by all pair-wise multiple comparisons), but had similar severity with the cheek group ($p > 0.05$, Kruskal–Wallis test followed by all pair-wise multiple comparisons).

*Secondary features*

The nasal group had the least severe burning/stinging and dry appearance compared with the other patient groups ($P < 0.05$ for all, Kruskal–Wallis test followed by all pair-wise multiple comparisons). Only 9% reported to have plaques, and there was no significant difference in severity or frequency of plaques among the four patient groups ($P > 0.05$ for both, Chi-square test and Kruskal–Wallis rank sum test). Edema was reported in 4.6% patients with more severe in the perioral group or cheek group than the full-face group ($p = 0.043$ and $p = 0.029$, Kruskal–Wallis test followed by all pair-wise multiple comparisons).

Phymatous changes was not so common, as only 51 patients were affected, all occurring in the nasal area, with 48 (94.1%) of them in the nasal group and three (5.9%) in the full-face group. Patients in the nasal group presented with phymatous changes as the following severity distribution: absent (7.7%), mild (44.2%), moderate (34.6%), and severe (13.5%). Patients in the full-face group presented with phymatous changes with a severity distribution of absent (99.4%), mild (0.6%), and moderate and severe (0.0%). Phymatous changes were much more severe in the nosal group than in the full-face group ($P < 0.001$, Wilcoxon test).

## Transitions between the groups

All the patients in the perioral group had their rosacea lesions start and remain on the chin. In the nasal group, patients all had their first lesions in the nose where it continues to remain, and most develop clinically apparent phymatous changes when admitted. In the full-face group, 55.8% of the patients had their lesions start on the full-face, 40.1% on the cheek, and only 4.1% on the nose. For those with lesions that started on the nose, most of them remain erythematotelangiectatic or papulopustules as they presented in the beginning, and only three cases developed mild phymatous change with no clinically apparent hypertrophic rhinophyma in a mean of 11.2 years.

## DISCUSSION

Although the National Rosacea Society Expert Committee (NRSEC) diagnosis and classification system has been incorporated in many basic, clinical, and epidemiological investigations (*Bae et al., 2009*; *Abram et al., 2010*; *Abram, Silm & Oona, 2010*; *Aksoy et al., 2010*; *Khaled et al., 2010*; *Lazaridou et al., 2010*; *Tan et al., 2013*), shortcomings of the NRSEC recommendations do exist, which does not make it universally accepted (*Wilkin et al., 2002*; *Crawford, Pelle & James, 2004*; *Van Zuuren et al., 2015*). The present descriptive study design evaluated the differences in clinical features among rosacea patients with lesions at four different sites for the first time.

A dysfunction of the epidermal barrier was observed in rosacea patients in the previous studies (*Dirschka, Tronnier & Folster-Holst, 2004*; *Lee, Jeong & Ahn, 2006*; *Ramos-e Silva*

& Jacques, 2012); we detected the barrier-associated physiological indicators that showed lower water content, higher TEWL, and hemoglobin in our rosacea patients compared with the healthy control. An interesting result of our study was the epidermal barrier function of the cheeks and chin in the nasal group was close to normal, which might explain, at least partly, why the nasal group had the least amount of burning, dryness, itching and overall symptoms. On the other hand, the more severe self-report symptoms observed in the full face or cheeks or perioral groups might be attributed to the impaired epidermal barrier. These distinctive features might be associated with the distinct physiological characteristics of local facial skin. In previous studies, the symptoms of burning/stinging were compared between patients with erythematotelangiectatic rosacea or papulopustular rosacea. However, no consistent conclusions were reached (*Lonne-Rahm, Fischer & Berg, 1999*; *Crawford, Pelle & James, 2004*). We inferred that the inconsistent results of these studies might have been attributed to the differences of the affected areas of the patients included. These findings need to be further confirmed, but if it is the different sites involved that contribute to the epidermal barrier functions and the clinical symptoms, the obtained conclusion may be beneficial for optimizing therapy schemes. For example, for patients who have full-face, cheek or perioral involvement, restoring the normal epidermal barrier function would be particularly important (*Bikowski, 2001*).

In this study, transitions between the four groups were addressed by surveying the sequence of the involved locations in rosacea patients by retrospective analysis. We found that (1) there were no transitions between the perioral groups and the other groups; (2) there were only a few patients in the full-face group whose lesions first occurred on the nose with the appearance of erythematotelangiectasia or papulopustule, and most of them keep the erythematotelangiectasia or papulopustule appearance on the nose; (3) Mild rhinophyma was occasionally seen in the full-face group, especially for those first lesions occurring in the nose, but moderate to severe rhinophyma, which means distortion of contour, only occurred in the nasal group, and transitions between the pre-existing rhinophyma in the nasal group with other groups were not observed; and (4) the cheek group could progress to a full-face. Considering the fact that the self-reported, dermatologist-evaluated grading of symptoms and physiological indicators of epidermal barrier function between the cheek and full-face groups did not have much difference, and the full-face group could have their lesions start at the cheek in majority of the patients, moreover, all the patients have lesions on the cheek when they were included in, we have speculated the cheek group might be a mild preceding form of the full-face group. However, the findings mentioned above require further investigation in a larger population with some following-up data.

Combined with the differences in the severity of symptoms, physiological indicators of epidermal barrier function, and disease outcomes between the nasal group and the other groups mentioned above, the following was observed: rosacea that occurred in the cheek or in the perioral area or in the full face were characterized by obvious self-conscious symptoms and abnormal epidermal barrier function, along with a relatively low risk of rhinophyma formation; thus, treatment should be focused on restoring the normal epidermal barrier function. On the contrary, rosacea with locations restricted to the nose meant milder symptoms and relatively normal epidermal barrier function but a potential

progression to rhinophyma. Early medical treatments then become indispensable to avoid irreversible disfiguring of the nose.

Consistent with previous studies, the majority of patients included in our study were women, but more than half of the nasal group were men, further confirming that men are at greater risk of having an affected nasal area (*Jansen & Plewig, 1997*; *Kyriakis et al., 2005*; *Abram et al., 2010*; *Spoendlin et al., 2012*; *Tuzun et al., 2014*). It might be associated with the different hormone levels or the facial skin physiology between women and men, while no related studies have been presented so far. As for the ocular subtypes of rosacea, because patients were recruited mainly in dermatologic clinics, patients with single ocular disorders were not included in our study, but 46 (7.9%) presented with ocular manifestations concomitantly, the prevalence of which is within the range of recent investigations (6–72%) (*Spoendlin et al., 2012*). Interestingly, all the patients with ocular symptoms were in the full-face group. It has been hypothesized that ocular rosacea tends to occur in patients with more extensive lesions on the face. Thus, careful ocular examination is recommended.

## CONCLUSIONS

In summary, herein, we revealed that rosacea lesions on the cheek or full face, nose and perioral of the face are associated with distinct features, including the proportion of male vs. female patients, self-reported symptoms, doctor-evaluated severity of symptoms and signs, and physiological indicators of epidermal barrier function. We also found that the localization of each group was relatively stable and barely transformed into other types. Our data may be beneficial for modifying the existing classification and stage definitions, optimizing clinical treatments, and facilitating mechanism researches of rosacea in the future.

## ACKNOWLEDGEMENTS

We thank Dr. Sai Zhang for her statistical help.

### Funding

This work was supported by the National Natural Science Foundation of China (No. 81502750). The funders had no role in study design, data collection and analysis, decision to publish, or preparation of the manuscript.

### Grant Disclosures

The following grant information was disclosed by the authors:
National Natural Science Foundation of China: 81502750.

### Competing Interests

The authors declare there are no competing interests.

## Author Contributions

- Hong-fu Xie conceived and designed the experiments, contributed reagents/materials/-analysis tools, reviewed drafts of the paper.
- Ying-xue Huang contributed reagents/materials/analysis tools, wrote the paper, prepared figures and/or tables.
- Lin He performed the experiments, analyzed the data, wrote the paper, prepared figures and/or tables.
- Sai Yang performed the experiments, prepared figures and/or tables.
- Yu-xuan Deng performed the experiments.
- Dan Jian and Wei Shi contributed reagents/materials/analysis tools.
- Ji Li conceived and designed the experiments, reviewed drafts of the paper.

## Human Ethics

The following information was supplied relating to ethical approvals (i.e., approving body and any reference numbers):

The ethics review board of XiangYa Hospital granted ethical approval to carry out the study within its facilities (Ethical Application Ref: 201212079).

## Data Availability

The raw data has been supplied as Data S1.

## Supplemental Information

Supplemental information for this article can be found online at http://dx.doi.org/10.7717/peerj.3527#supplemental-information.

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
