# Peer review of "An observational descriptive survey of rosacea in the Chinese population: clinical features based on the affected locations"

_PeerJ, doi:10.7717/peerj.3527_

## Round 0.1 · original submission · Minor Revisions

Dear Ji,

Thank you for your submission to PeerJ.

It is my opinion as the Academic Editor for your article - An observational descriptive survey of rosacea in the Chinese population: clinical features based on the affected locations - that it requires a number of Minor Revisions. Please address all the comments of the reviewers.

Reviewer 1 ·

Basic reporting

The authors reported the significant differences in clinical features among rosacea patients with lesions at four different sites . The manuscript were written in clear English.

Experimental design

no

Validity of the findings

no

Additional comments

1. What’s the constitution of the full-face group? The authors proposed the cheek group could progress to a full-face, and patients in 40.1% of the full-face group had their lesions start on the cheek, so I wonder if there is any patients in the full-face group with lesions on the facial area but not including the cheeks? what’s the proportion?
2. Please explain the difference of melanin contents among the four groups.
3. In the abstract, the authors mentioned the cheek group had lower water content in the chin compared with the healthy control, but the Results part and the tables did not show this result. Please give an explanation.
minor comments:
1.It is better to format the Tables into standard three line table.
2.In line 323, "All the authors declare that they have no conflict of Interest", it is not appropriate to show this statment in the section of Acknowledgements.

Reviewer 2 ·

Basic reporting

1. What is the innovation in this manuscript? Please discuss or explain in the text.
2. What is the impact of findings on clinical diagnose and treatment? Please detail the significance in the text.

Experimental design

1. Were the patients recruited from one hospital only? How about the others??
2. It should be the Conclusion instant of Discussion in Abstract (lines 19).
3. In Table1, it should be “Female and Male”, and the title should be moved prior to the Table.
4. In lines 147, a full name TEWL should be given at the first time in the text.
5. In lines 191-214, please add one column in the Table 4 for the total (%).
6. Please remove the first paragraph from lines 248-255 to introduction section.
7. The English should be improved to ensure that reader can understand the manuscript clearly. Please check the text with a native English speaker.
8. No page number in the text.

Validity of the findings

no comment

Additional comments

The manuscript submitted by Xie et al describes that rosacea lesions on the cheek or full face, nose and perioral of the face are associated with distinct features, including the proportion of male vs. female patients, self-reported symptoms, doctor-evaluated severity of symptoms and signs, and physiological indicators of epidermal barrier function. Also authors found that the localization of each group was relatively stable and barely transformed into other types. However, some of concerns are raised as follows.

Reviewer 3 ·

Basic reporting

No comment

Experimental design

No comment

Validity of the findings

No comment

Additional comments

We think your concept is interesting and potentially valuable. Your data showed that rosacea lesion locations are associated with several distinct features, like self-reported symptoms, physiological indicator of epidermal barrier function. However, what is the relationship between your rosacea lesion site classification with the existing classification/stage definitions? Are there any different clinical outcome in each rosacea lesion sites? How to guide the clinical treatment using the rosacea lesion site classification?

---

## Round 0.2 · accepted · Accept

I am pleased to inform you that your manuscript was accepted after careful review.

Reviewer 1 ·

Basic reporting

no

Experimental design

no

Validity of the findings

no

Additional comments

no

Reviewer 3 ·

Basic reporting

no comment

Experimental design

no comment

Validity of the findings

no comment